# In Silico Structural and Biochemical Functional Analysis of a Novel *CYP21A2* Pathogenic Variant

**DOI:** 10.3390/ijms21165857

**Published:** 2020-08-14

**Authors:** Michal Cohen, Emanuele Pignatti, Monica Dines, Adi Mory, Nina Ekhilevitch, Rachel Kolodny, Christa E. Flück, Dov Tiosano

**Affiliations:** 1Pediatric Endocrinology Unit, Ruth Rappaport Children’s Hospital, Rambam Healthcare Campus, Haifa 352540, Israel; d_tiosano@rambam.health.gov.il; 2The Ruth and Bruce Rappaport Faculty of Medicine, Technion, Israel Institute of Technology, Haifa 352540, Israel; 3Pediatric Endocrinology, Diabetology and Metabolism, Department of Pediatrics, University Hospital Inselspital, University of Bern, 3010 Bern, Switzerland; emanuele.pignatti@dbmr.unibe.ch (E.P.); Christa.Flueck@insel.ch (C.E.F.); 4Department of BioMedical Research, University Hospital Inselspital, University of Bern, 3010 Bern, Switzerland; 5Sagol Department of Neurobiology, University of Haifa, Mount Carmel, Haifa 31905, Israel; monica.dines@gmail.com; 6Genetics Institute, Rambam Health Care Campus, Haifa 3525408, Israel; A_Mory@rambam.health.gov.il (A.M.); n_ekhilevitch@rambam.health.gov.il (N.E.); 7Department of Computer Science, University of Haifa, Mount Carmel, Haifa 3498838, Israel; trachel@gmail.com

**Keywords:** congenital adrenal hyperplasia, *CYP21A2*, *in silico*, protein structure, VUS, consurf

## Abstract

Classical congenital adrenal hyperplasia (CAH) caused by pathogenic variants in the steroid 21-hydroxylase gene (*CYP21A2*) is a severe life-threatening condition. We present a detailed investigation of the molecular and functional characteristics of a novel pathogenic variant in this gene. The patient, 46 XX newborn, was diagnosed with classical salt wasting CAH in the neonatal period after initially presenting with ambiguous genitalia. Multiplex ligation-dependent probe analysis demonstrated a full deletion of the paternal *CYP21A2* gene, and Sanger sequencing revealed a novel de novo *CYP21A2* variant c.694–696del (*E232del*) in the other allele. This variant resulted in the deletion of a non-conserved single amino acid, and its functional relevance was initially undetermined. We used both in silico and in vitro methods to determine the mechanistic significance of this mutation. Computational analysis relied on the solved structure of the protein (Protein-data-bank ID 4Y8W), structure prediction of the mutated protein, evolutionary analysis, and manual inspection. We predicted impaired stability and functionality of the protein due to a rotatory disposition of amino acids in positions downstream of the deletion. In vitro biochemical evaluation of enzymatic activity supported these predictions, demonstrating reduced protein levels to 22% compared to the wild-type form and decreased hydroxylase activity to 1–4%. This case demonstrates the potential of combining in-silico analysis based on evolutionary information and structure prediction with biochemical studies. This approach can be used to investigate other genetic variants to understand their potential effects.

## 1. Introduction

Congenital adrenal hyperplasia (CAH), one of the most frequent inborn errors of metabolism, comprises a group of autosomal recessive disorders of adrenal steroidogenesis (OMIM 201910). Abnormalities in the steroid 21-hydroxylase account for the majority of cases [1]. Impaired function of this enzyme results in varying degrees of cortisol and aldosterone deficiencies and consequently increased production of the adrenocorticotropic hormone (ACTH). This leads to chronic over-stimulation of the adrenal gland and increased synthesis of androgens starting in the first weeks of gestation. Steroid 21-hydroxylase deficiency has a broad spectrum of clinical presentations ranging from severe, classical CAH that presents early after birth to the mild late-onset form (also referred to as non-classical CAH) [2]. Neonatal screening programs report the incidence of classical CAH as being approximately 1:15,000 live births [3]. The incidence of the non-classical form is much higher at about 1:200–1:1000 worldwide. There are clear genotype–phenotype correlations reflecting the 21-hydroxylase residual enzymatic activity. Mutations leading to complete inactivation of steroid 21-hydroxylase are usually associated with a classical CAH phenotype, whereas those with significant residual enzymatic activity usually present with a non-classical CAH phenotype [2]. The steroid 21-hydroxylase enzyme is a member of the cytochrome P450 group of oxidases [4]. These enzymes contain a single heme group that absorbs light with a wavelength maximum at 450 nm in their reduced monoxide-bound states. The steroid 21-hydroxylase consists of 495 amino acids and is localized in the membrane of the endoplasmic reticulum facing the cytosol where it catalyzes the 21-hydroxylation of two natural substrates, progesterone and 17-hydroxyprogesterone [2,4,5,6]. The protein structure of human steroid 21-hydroxylase was solved (Protein Databank ID 4y8w), using X-ray crystallography, in the presence of the heme and progesterone or 17-hydroxyprogesterone [7,8]. This protein has 13 alpha helices and 9 beta strands, all organized around the heme.

The steroid 21-hydroxylase gene *CYP21A2* is located on the short arm of chromosome 6 (6p21.3) within the human leukocyte antigen complex and it includes 10 exons. It is adjacent to the pseudogene *CYP21A1P*, with which it shares 98% sequence identity [1,5]. The *CYP21A2* gene has been characterized and studied in detail, and multiple genetic lesions have been published [9]. Most of the described disease-causing variants are likely to be the consequence of non-homologous recombination or gene conversion events between the gene and the pseudogene [5,10,11,12].

We describe a novel *CYP21A2* variant causing in-frame deletion of a single amino acid found in a compound heterozygote state with a larger deletion comprising the whole *CYP21A2* gene. The patient had severe classical salt-wasting CAH. We present an extensive study of this new mutation and its effect on both the structure and function of the enzyme. Based on the in-silico analysis of the three-dimensional molecular model for CYP21A2 and the in vitro biochemical evaluation of the enzymatic activity, we provide a structural explanation and characterization of the enzymatic dysfunction.

## 2. Results

### 2.1. Case Report

The patient was born at 36 weeks’ gestation after premature spontaneous rupture of membranes, and her birth weight was 2415 g (0.12 weight z-score). The pregnancy was spontaneous, and the results of routine gestational ultrasound examinations were normal. Amniocentesis was performed due to advanced maternal age (38 years) and it revealed a 46 XX karyotype. The parents are of Ashkenazi-Jewish origin, with no consanguinity, and no known history of CAH in the family. Physical examination at birth revealed signs compatible with classical salt wasting CAH, including clitoromegaly, fused labial folds with a single urogenital opening at the base of the phallus, and no palpable gonads, in accordance with stage 3 on the Prader scale [13]. Laboratory tests on the 2nd day of life and during an episode of significant respiratory distress demonstrated an elevated testosterone level of 30 nmol/L (normal values 0.7–2.2 nmol/L) a 17OHP level of 27 nmol/L (normal range 2–70 nmol/L) and a relatively low cortisol level of 152 nmol/L. Abdominal ultrasound demonstrated bilaterally enlarged adrenal glands. She was clinically suspected of having CAH and hydrocortisone treatment was initiated. Hyponatremia and hyperkalemia with urinary salt wasting were noted at 10 days of age, and treatment with fludrocortisone and salt supplementation was added. Cystoscopy at 1 month of age demonstrated a 2-cm urogenital tunnel. Two surgical procedures were performed: she underwent meatoplasty at 12 months of age and a feminizing genitoplasty at 2.5 years of age. Throughout infancy and childhood, standard doses of hydrocortisone and fludrocortisone were required for control of her disease, although the 17OHP levels were increased up to >60 nmol/L (normal range 2–6 nmol/L) on several occasions during her follow-up. Salt supplementation was gradually tapered after the 2nd year of life. The patient is growing well and her neurological development is normal.

### 2.2. DNA Studies

MLPA analysis demonstrated a heterozygote deletion that included the *CYP21A2* gene, and this *CYP21A2* deletion was detected in the paternal sample as well. Sequencing of the *CYP21A2* (NM_000500.7) gene revealed an additional *CYP21A2* in-frame deletion of 3 nucleotides encoding glutamic acid, c.694_696del; *E232del* (p.Glu232del). This variant was absent in both parents’ sequencing analysis, and therefore assumed to be de novo. It also had not been previously described and its functional relevance was undetermined.

### 2.3. In Silico Analysis of Protein Structure and Function

ConSurf [14,15,16,17,18] analysis of CYP21A2 and its homologues demonstrated evolutionarily conserved residues of structural and functional importance. We used the ConSurf server for the pdb structure 4y8wA, employing default parameters (150 homologues). Figure 1A shows the relevant part of the structure that was colored by evolutionary conservation. The evolutionary conservation scale in ConSurf ranges from 1 (non-conserved, colored in cyan) to 9 (highly conserved, colored in burgundy). Generally speaking, helix G (following the helix-naming scheme of Pallan et al. [7]), which hosts E232, is peripheral and mostly variable. Of relevance here, all of the residues in this helix that face the solvent, and E232 in particular, are variable (namely, the residues Q228, E232, K233, H236, E239, and M240). Indeed, a mutation of M240K was observed, and it was reported to be non-harmful [19]. E232, which is colored in cyan to indicate that it is not evolutionarily conserved, is remote from the substrate, and its side chain faces outward towards the solvent. We predicted the deletion of E232 to have an impact on the residues within the same helix (helix G). There are 3.6 residues per alpha-helix turn, therefore, the impact of the deletion is to shorten the helix by approximately 1/3 of a turn, and to alter the position of each of the downstream residues by turning it back or rotating it (by approximately 1/3 of a turn).

This structural change is expected to impair protein stability. Helix G on the surface of the protein faces the buried helix I. The side chains of the residues R234, D235, and V238 in helix G face the side chains of the residue H283 in the helix I, and their shapes are complementary with the H positioned in a groove formed by the R, D, and V. This positions helix G properly in place. The electrostatics also contribute to securing the position of helix G, with the positive charge of the H283 fitting into the negatively charged groove. Indeed, these four residues are very well conserved (R234, D235, and V238 have the maximal consurf score of 9, and H283 has that consurf score of 8). Thus, we suggest that the deletion of the residue E232 is harmful to the stability of the protein because of its impact on the R234, D235, and V238 located 2, 3, and 6 residues ahead in the polypeptide chain. The deletion of the residue at the beginning of helix G, and the rotation outwards of these residues (R234, D235) likely disrupts the packing of the two helices against each other, disrupting the protein’s overall fold.

This structural change has further significance due to its impact on residue R234, which participates in positioning the CYP21A2 ligand, and thus has a particularly important role in the function of the protein. In the native structure (4y8w), the hydrogen at the tail of the R234 side chain hydrogen bonds with the progesterone’s O3, helping to position the progesterone so that its carbon atom C21 on the other end lies near the heme iron (Figure 1B). The rotameric state of residue R234 is very similar among the homologues of known structures. Figure 1C shows the three PDB homologues: 4y8w (human, pink), 3qz1 (bovine, gray), and 6b82 (zebra fish, gray). In the insert, the three side chains are seen to be superimposed, such that the hydrogen is in the same spatial position, i.e., facing inward, where it can interact with the progesterone. Finally, in support of the functional importance of R234, previous studies have documented mutations with impaired function in this residue. Barbaro et al. [21] described two patients with an R234G mutation manifested as a non-classical CAH phenotype, and Tardy et al. [22] measured the relative activity of an R234K mutation in vitro and showed that it is comparable to the I231T mutant (which also manifests as non-classical CAH [19]). The deletion leads to the rotation of R234 outwards, toward the solvent, rather than facing and interacting with the progesterone molecule. Indeed, when computationally predicting the structure with E232 having been deleted, the Phyre2 web-server [20] demonstrates this precisely. Such a rotation in the positioning of the R234 is expected to significantly impair the enzyme’s ability to hydrolase its substrate. We followed this in silico analysis with in vitro studies.

### 2.4. In Vitro Biochemical Evaluation of Enzymatic Activity

To assess the impact of the novel *E232del* variant on the steroid 21-hydroxylase activity, we expressed the wild type (WT) and *E232del CYP21A2* in non-steroidogenic COS1 cells and quantified the hydroxylase activity for both the WT and variant proteins. Using thin layer chromatography, we found that this mutation negatively affects the hydroxylase activity of steroid 21-hydroxylase for both progesterone and 17-OH progesterone substrates compared to the WT-CYP21A2 (4% and 1%, respectively) (Figure 2A). To determine whether the reduced activity of *E232del*-CYP21A2 is associated with decreased expression of the protein, we quantified both *E232del*-CYP21A2 and WT-CYP21A2 expression using Western blot analysis. We found that the *E232del* mutation reduces the expression of steroid 21-hydroxylase to 22% compared to the WT form (Figure 2B). We calculated the specific activity of *E232del*-CYP21A2 compared to the WT-CYP21A2, by dividing the activity obtained by TLC by the protein level. The specific activity of *E232del*-CYP21A2 was decreased to 16% compared to the WT protein. Altogether, our results indicated that the E232del mutation significantly impacted both the expression and activity of steroid 21-hydroxylase.

## 3. Discussion

We present a detailed mechanistic investigation of the molecular phenotype of a novel de novo in-frame deletion *c.694_696del* that was found in a compound heterozygote state with a larger deletion comprising the whole *CYP21A2* gene, in a patient diagnosed as having classical salt-wasting CAH.

The *E232del* variant results in the deletion of a non-conserved amino acid (glutamic acid at position 232) and thus was initially interpreted to be a variant of uncertain significance. CAH is a serious life-threatening condition. Molecular diagnosis can be crucial for families, since it may affect counseling and future pregnancy planning. In recent years, a surge of clinically directed genetic tests has become available, dramatically increasing diagnostic yield and improving medical management and genetic counselling [23,24,25,26,27,28,29,30]. However, these new opportunities bear significant challenges. One major challenge is interpreting and analyzing rare sequence variants that are not necessarily clinically relevant [31,32]. In particular, the impact of changes, such as missense and in-frame indel variants might be more difficult to interpret. On average, 2% of the population carry a missense variant in any given gene, emphasizing the extent of this difficulty [33]. Similar to the result in our case, even when a new variant is highly suspicious of being pathogenic, without functional reasoning, pathogenicity remains uncertain [27] and additional investigation methods are required.

In vitro biochemical assays are key to the study of novel gene variants and the results in the presented case are in agreement with the clinical presentation of classical salt-wasting CAH. However, such investigations are time- and resource-consuming, emphasizing the importance of correctly choosing the relevant functional assay. Different assays might be used based on the particular aspect of the protein’s function under assessment. For example, assays that characterize variants interfering with an enzyme’s active site differ from those directed at evaluating its rate of degradation [34,35]. Thus, to better understand specific variants in patients and to plan the functional assays, one needs a mechanistic understanding of the impact of the variant. To this end, in-silico approaches can be extremely helpful.

We used computational methods to determine the effect of the gene variant on the steroid 21 hydroxylase enzyme. Our analysis findings suggest that the variant affects the intra-helical positioning of amino acids downstream from the deleted residue. This significantly impairs the proteins’ electrostatic stability and enzymatic function. Structural models can explain not only missense variants, but also more complicated cases, such as deletions of a single amino acid, as demonstrated in our patient. Accurate crystallography-based analysis of the protein structure is critical to a mechanistic model [36,37,38,39]. Based on our analysis, we further predict that missense mutations among variants in the *CYP21A2* gene residues in the helix G preceding the R234, starting at residue 225 and facing the solvent (R225 R226 Q229 A230 E232 K233), will not be very harmful, but that deletions and perhaps insertions will disrupt the function of the protein.

In conclusion, *c.694_696del* is a novel variant of the *CYP21A2* gene causing classical salt-wasting CAH, broadening the spectrum of known mutations related with CAH. This case demonstrates the mechanistic insights that can be gained from a comprehensive computational analysis based on evolutionary information and structure prediction. We believe that the large numbers of possible genetic variants and their potential effects dictate the implementation of computational tools to support clinicians and scientists in the interpretation of genetic test results.

## 4. Materials and Methods

### 4.1. Genetic Analysis

Genetic testing was performed on clinical grounds with the approval of the Rambam Healthcare Campus institutional ethics board. Tests were performed on peripheral blood-derived DNA samples from the patient and both of her parents who gave their written informed consent for testing and publication. Multiplex ligation-dependent probe (MLPA) analysis (SALSA MLPA Probemix P050-C1 kit, MRC Holland, Amsterdam, The Netherlands) was performed to detect gene deletions and duplications. Sanger sequencing was performed for the entire *CYP21A2* gene by means of two-step PCR amplification to specifically amplify the *CYP21A2* gene, as previously described [40]. For gene sequencing, capillary electrophoresis was implemented using the 3500 genetic analyzer. The results were analyzed with Sequencher v5.0 (Gene Codes Corporation, Ann Arbor, MI, USA).

### 4.2. In Silico Studies: Mechanistic Analysis of Protein Function with a Three-Dimensional Molecular CYP21A2 Model

The structure of the human CYP21A2 protein complexed with progesterone or with 17-hydroxyprogesterone was solved by Egli, Guengeric and co-workers, and it was deposited in the Protein DataBank (ID 4Y8W) [7,8]. To understand the significance of the molecular phenotype of this mutation, we relied on evolutionary data for comparing homologous proteins [36]. We used the ConSurf suite [14,15,16,17,18] to deduce the effect of the mutation on the structure and function of protein residues based on their evolutionary conservation in homologous proteins. We mapped this information back to the three-dimensional structure of the protein and manually assessed the effect on the proteins’ structure and enzymatic activity.

### 4.3. Biochemical Evaluation of Enzymatic Activity

#### 4.3.1. Cell Transfection and Steroid Profiling

COS1 cells were transfected with 3 μg of mammalian expression plasmid (pcDNA3.1 (+), Addgene) containing the full coding sequence of WT-*CYP21A2* or *E232del*-*CYP21A2* genes (GenScript). Both coding sequences were flanked by a DYK (FLAG) tag expressed at the C terminal of the protein. Experiments were performed in 12-well plates (3 × 10^5^ cells/well). Transfection was performed with Lipofectamine 2000 (Thermo Fisher, Waltham, MA, USA) for 4 h, after which the growth media was replaced. At 48 h after transfection, new media were added containing both 80,000 cpm of [H^3^]-progesterone and [H3]-17-OH-progesterone, corresponding to a substrate about 15 nM. The medium was collected 90 min after. The amount of radioactive tracer steroids and time of incubation were chosen based on previously published work that established the ability to convert progesterone to 11-deoxycorticosterone and 17-OH-progesterone to 11-deoxycortisol in a cell-based system [41]. Steroids were extracted and separated by thin layer chromatography (TLC) as previously described. Relative steroid conversion was quantified as a percentage of incorporated radioactivity into 11-deoxycorticosterone and 11-deoxyxortisol in relation to total radioactivity measured in the whole sample and compared between the WT-CYP21A2 and the CYP21A2 variants.

#### 4.3.2. Western Blot

For protein analysis, transfected cells were washed with ice-cold PBS and harvested in lysis buffer containing 14.6 mg EDTA, 121 mg trisaminomethane (Tris), 435 mg NaCl, 0.5 mL Triton X-100 (Thermo Fisher) and cOmplete^TM^ protease inhibitor cocktail (Roche, Basel, Switzerland) in 50 mL of water. Seven μg of total protein were then loaded on a precast gradient gel (GenScript, Piscataway, NJ, USA, Cat# M81615). Proteins were transferred on a PVDF membrane at 4 °C overnight and blocked with 5% non-fat milk in tris-buffered saline (TBS) supplemented with 0.1% Tween-20 (TBST). The following primary antibodies were used: mouse anti-FLAG (GenScript, Cat# A00187), diluted 1:1000, and rabbit anti-GAPDH (Santa Cruz, CA, USA Cat# sc-25778), diluted 1:1500. Incubation with primary antibodies was performed in 1% non-fat milk in TBST, overnight at 4 °C with mild rocking. The following secondary antibodies were used: IRDye 680RD-conjugated donkey-anti-mouse (LI-COR, Lincoln, NE, USA, Cat# 926-86072), diluted 1:10,000, and IRDye 800CW-conjugated donkey-anti-rabbit (LI-COR, Cat# 926-32213), diluted 1:5000. Incubation with secondary antibodies was performed in 1% non-fat milk in TBST at room temperature for 1 h with mild rocking. Membranes were washed with TBST, and signals were detected by the Odyssey^®^ Sa infrared Imaging system (LI-COR Bioscience Inc.).

#### 4.3.3. Statistics

Statistical significance was calculated with a two-tailed Student’s *t*-test for comparing two groups. For every two-group comparison, we ran an *F*-test to assess whether those two groups had different variances. If there were a different variance, the two-tailed Welch *t*-test was used instead of the Student’s *t*-test. Graphs with more than two groups were analyzed by two-way ANOVA analysis followed by post hoc Sidak’s multiple comparison test. Prism 8 (Graphpad, San Diego, CA, USA) and Excel (Microsoft, Redmond, WA, USA) were used for the statistical analyses and calculations.

Informed consent was obtained from the patient’s guardian for publication of the case report. The study was presented to our institutional review board (IRB) and approval was waived based on the clinical nature of the investigations performed.

## Figures and Tables

**Figure 1 ijms-21-05857-f001:**
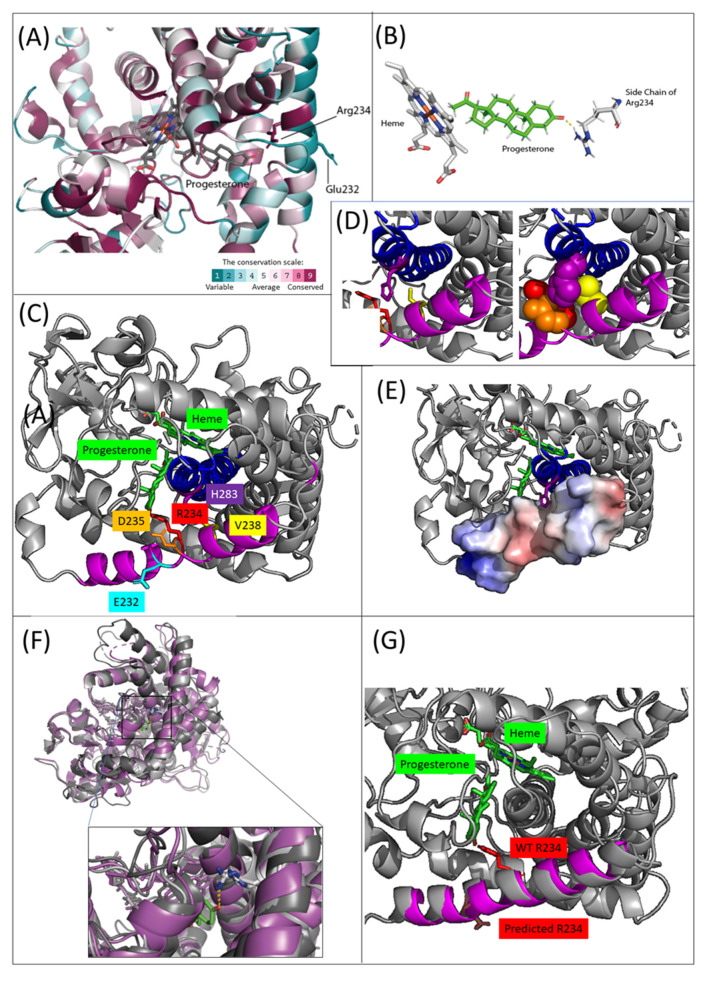
Highlighting evolutionary and structural information relevant to our case in the CYP21A2 structure. Legend: (**A**): the relevant part of the CYP21A2 structure colored by evolutionary conservation. The evolutionary conservation scale in ConSurf ranges from 1 (non-conserved, colored in cyan) to 9 (highly conserved, colored in burgundy). (**B**): The native CYP21A2 structure (4y8w), in which the hydrogen at the tail of the R234 side chain hydrogen-bonds with the progesterone’s O3, helping position the progesterone so that its carbon atom C21 on the other end lies near the heme. (**C**): Highlighting several structural elements: helices G and I are shown in magenta and blue, and residues E232, R234, D235, V238, and H283 are highlighted in different colors, with their sidechains shown as sticks. (**D**): Zooming in on the region where helix G and helix I touch each other, and showing the side chains of the relevant residues (as sticks—left panel, and spheres—right panel), the H283 (in purple) is seen to be positioned in the groove formed by the R, D, and V. The distances between the ring of H283 and the side chains of these residues are 4.1A (V238), 4.8A (D235), and 5.9A (R234). (**E**): The electrostatic potential of helix G (ranging from red for negatively charged to blue for positively charged) shows the groove that holds H283. (**F**): The side chains of the three PDB homologues (3qz1, a bovine homologue and 6b82 a zebra fish homologue) of the P45021A2 are shown superimposed. (**G**): The predicted structure by Phyre2 [20]. For clarity of representation, we show only helix G in magenta, R234 in red, and the bound substrates from the WT structure; the predicted R234 is facing away from the progesterone.

**Figure 2 ijms-21-05857-f002:**
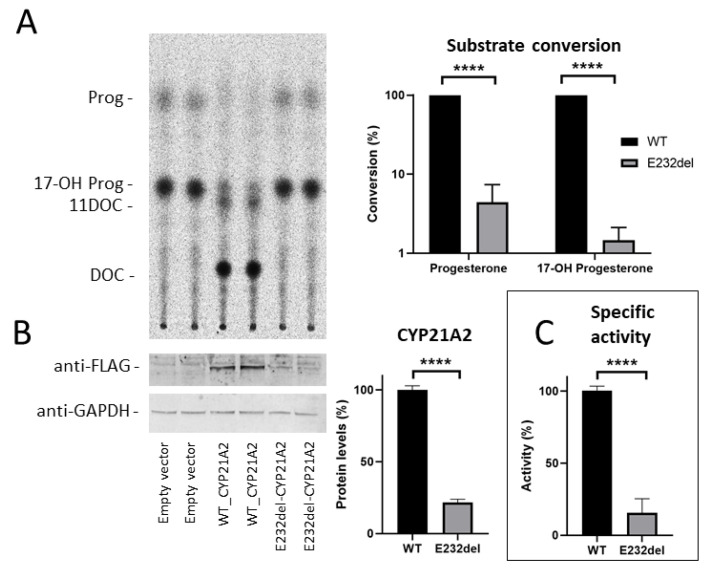
CYP21A2-*E232del* affects protein levels and 21-hydroxylase activity. Legend: (**A**) Left-hand panel: thin layer chromatography (TLC) shows that *E232del* variant markedly reduces the hydroxylase activity of CYP21A2. Right-panel: substrate conversion is calculated after densitometric quantification of the relative incorporation of radioactivity in specific TLC spots. Conversion of progesterone (Prog) to 11-deoxycorticosterone (11DOC) in the mutated protein (*E232del*-CYP21A2) is reduced to an average of 4% compared to the wild-type form of the protein (WT-CYP21A2). Conversion of 17-hydroxyprogesterone (17-OH Prog) to 11-deoxycortisol (DOC) is reduced to an average of 1%. (**B**) Left panel: Western blot analysis of protein levels following overexpression of CYP21A2 in non-steroidogenic COS1 cells shows reduced expression of the protein carrying the *E232del* variant compared to the wild-type form. Right panel: densitometric quantification of the blot indicating that *E232del* mutation reduces CYP21A2 expression to 22% on average. (**C**) Relative specific activity was calculated by dividing the values of substrate conversion (Panel A) by the paired protein level values (Panel B). The *E232del* variant then reduces CYP21A2-mediated activity to 16% on average. Statistical analysis was conducted as described in the Methods section. The bars in Panel A were analyzed using a two-way ANOVA analysis followed by post hoc Sidak’s multiple comparison test. The bars in Panel B were analyzed using a two-tailed Student’s *t*-test. The bars in Panel C were analyzed using a two-tailed Welch *t*-test. The error bars indicate the standard error of the mean (SEM). Every experiment was run with three independent biological replicates, each time using technical triplicates. ****, *p* ≤ 0.0001.

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
