# Peer review of "In Silico Structural and Biochemical Functional Analysis of a Novel CYP21A2 Pathogenic Variant"

_ijms, 2020, doi:10.3390/ijms21165857_

Round 1

Reviewer 1 Report

The authors detected a novel p.Glu232del variant in the CYP21A2 gene. They performed both in silico and in vitro characterization and firmly concluded that it is pathogenic.

The manuscript will be important for genetic counselling and prenatal testing of families with this variant.

Author Response

Thank you very much for the review.

Reviewer 2 Report

The manuscript describes the characterisation of a newly discovered mutant of steroid 21-hydroxylase CYP21A2 using an in-vitro enzymatic assay and computational tools for the prediction of structural consequences of the mutation. In combination the results deliver a comprehensive assessment of the structure/function-relationship of the novel CYP21A2 mutant. However, the discussion of the structural impact of the mutation requires a more detailed description and illustration of the predicted structure of the mutant in comparison to the wildtype enzyme prior to discussing the functional impact of changes in the orientation of individual residues. The replicate numbers and choice of conditions for the enzymatic activity assay need clarification to demonstrate validity.

Major comments

Line 76 “Genetic testing was performed with the approval of the institutional ethics board”. Provide the details. Which institute?

Line 99 “80,000 cpm of [H3]-progesterone and [H3]-17-OH-progesterone were added to the culture medium…” and section 3.4: It needs to be clarified, how the incubations conditions of the enzymatic assay were selected and if they were suitable to compare the wildtype and mutant. What molar substrate concentrations does this correspond to and how does this compare to published Km values of CYP21A1? Were both substrates added at the same time or in separate incubations? Why was 90 minutes selected as incubation time?

Line 176-187: This section is difficult to follow, because the residues discussed are not indicated in any of the figures. I suggest add a panel to figure 1 highlighting the interactions between helices 10 and 12 as discussed in the text.

Line 202 “Indeed, when computationally predicting the structure with E232 deleted,

203 using the Phyre2 web-server (24), we see exactly this.”: This is an important finding and should be discussed at the beginning of the section. The structure/function consequences of E232del discussed in lines 176-205 are based on the hypothesis that the deletion leads to a rotation of the downstream amino acids in the helix. Hence, it should be shown first that this hypothesis is valid by introducing the model of the predicted structure of the variant. I suggest adding a figure comparing the crystal structure of the wildtype and the predicted structure of the new variant.

The enzymatic characterisation of the novel mutant compared to the wildtype is very brief (one undefined substrate concentration and one timepoint only). In the light of the negligible activity of the mutant under the condition tested, a full kinetic characterisation would not alter or strengthen the message of the manuscript. However, given the rough characterisation of the enzyme activity, the term “catalytic activity”, which is usually absolutely quantified as product per time under a given condition, should not be used. I suggest using (relative) substrate conversion or (relative) activity as appropriate.

The number of replicates for the activity assay are not clear (Figure 2). The number of biological replicates (and technical replicates, if applicable) should be indicated. What type of error is indicated by the error bars in Figure 2?  

Minor comments

Abstract “demonstrated a heterozygote deletion that includes the paternal CYP21A2 gene and Sanger sequencing revealed a novel de novo CYP21A2 variant c.694-696del (E232del).”: Rephrase. It does not become clear what the genetic status is and that there is a full deletion of the other allele

Abstract “reduced protein expression to 22% compared to the wild-type form”: What does this refer to? Protein levels in vivo or in the recombinant expression system?

Line 53: Please add references.

Line 55 “that absorbs light at 450nm in their reduced states”: This is not correct. It is in the reduced CO-bound state that the maximum is at 450 nm.

Line 56 “and is localized to the endoplasmic reticulum” Please rephrase the clarify that it is not in the ER, but localised in the membrane of the ER facing the cytosol

Line 95 “COS1 cells were transfected with 3μg of mammalian expression plasmid…” What well/culture dish size was used?

Line 106/107: Please provide details of the lysis buffer, especially detergents used to solubilise the P450.

Line 119 “…for groups of two”: Does this mean when comparing 2 groups? Please rephrase.

Line 138: Please provide reference ranges for the measurements.

Line 155: What is P450A2? Which homologues were used to conclude on evolutionary conservation?

Line 175: Which are the three PDB homologues?

Line 177: What is the distance between H283 and the residues of helix 10?

Line 192: use “heme iron”

Figure 2 and figure legend: Each figure/graph should have its own label A-E. Currently, it is not clear what C refers to.

Section 3.4 and figure 2: The use of the term “protein expression” should be reconsidered. What the authors have studied here using Western Blot are the levels of the protein in the cell, which result from protein biosynthesis/expression and protein degradation. No information can be deduced, whether there is less of the mutant protein is made (compared to the wildtype) or there is higher turnover/degradation (or both). To avoid misinterpretation, I suggest to use the term “protein levels”.

Check the use of “decreased to” vs. “decreased by” in section 3.4 and the legend of Figure 2.

Line 262: “We found the variant to affect the intra-helical positioning of amino acids downstream from the deleted residue.”: This should be rephrased to clarify that this based on a model of the predicted structure of the mutant and not based on direct evidence from structural studies.

Line 273: “In light of this difficulty in predicting significant structural changes caused by minor sequence changes, manually combining data from different computational tools was required in our case.” This seems to be contradicting a previous statement about the generation of a structure model for the mutant. Line 202: “Indeed, when computationally predicting the structure with E232 deleted, using the Phyre2 web-server (24), we see exactly this.” This should be clarified. Which different computational tools do they refer to? The method section only introduced the tools available on ConSurf.
